# Cytotoxic and Genotoxic Evaluation of the Aqueous and Hydroalcoholic Leaf and Bark Extracts of *Crataegus oxyacantha* in Murine Model

**DOI:** 10.3390/plants10102217

**Published:** 2021-10-19

**Authors:** Fany Renata Aguilera-Rodríguez, Ana Lourdes Zamora-Perez, Clara Luz Galván-Moreno, Rosalinda Gutiérrez-Hernández, Claudia Araceli Reyes Estrada, Edgar L. Esparza-Ibarra, Blanca Patricia Lazalde-Ramos

**Affiliations:** 1Maestría en Ciencia y Tecnología Química, Laboratorio de Etnofarmacología, Unidad Académica de Ciencias Químicas, Universidad Autónoma de Zacatecas, 98000 Zacatecas, Mexico; aguilerarenata468@gmail.com (F.R.A.-R.); clara.galvanmoreno@gmail.com (C.L.G.-M.); 2Instituto de Investigación en Odontología, Centro Universitario de Ciencias de la Salud, Universidad de Guadalajara, 44100 Jalisco, Mexico; anazamora@gmail.com; 3Programa de Licenciatura en Nutrición de la Unidad Académica de Enfermería, Universidad Autónoma de Zacatecas, 98000 Zacatecas, Mexico; rosalinda@uaz.edu.mx; 4Maestría en Ciencias de la Salud Unidad Académica de Medicina Human, Unidad Académica de Ciencias Biológicas, Universidad Autónoma de Zacatecas, 98000 Zacatecas, Mexico; c_reyes13@uaz.edu.mx; 5Unidad Académica de Ciencias Químicas, Universidad Autónoma de Zacatecas, 98000 Zacatecas, Mexico; edgarzac@gmail.com

**Keywords:** *Crataegus oxyacantha*, micronuclei, DNA damage, genotoxic, cytotoxic

## Abstract

*Crataegus oxyacantha* has been mainly used in traditional medicine for the treatment of cardiovascular diseases. However, its safety profile has not been fully established, since only the genotoxic effects of *C. oxyacantha* fruit have been described. Therefore, the objective of this work was evaluating the cytotoxicity and genotoxicity of the aqueous and hydroalcoholic leaf and bark extracts of *C. oxyacantha* by means of the micronucleus test in a murine model. Doses of 2000, 1000, and 500 mg/kg of both extracts were administered orally for 5 days in mice of the Balb-C strain. Peripheral blood smears were performed at 0, 24, 48, 72, and 96 h after each administration. The number of polychromatic erythrocytes (PCEs), micronucleated polychromatic erythrocytes (MNPCEs), and micronucleated erythrocytes (MNEs) was determined at the different sampling times. Our results showed that the leaf and bark of *C. oxyacantha* increase the number of MNEs at the 2000 mg/kg dose, and only the aqueous leaf extract decreases the number of PCEs at the same dose. Therefore, the aqueous and hydroalcoholic leaf and bark extracts of *C. oxyacantha* showed genotoxic effects, and only the aqueous leaf extract exhibited cytotoxic effects.

## 1. Introduction

*Crataegus oxyacantha* L. (Rosaceae) is a deciduous tree commonly known as white hawthorn and is a member of Rosaceae family. It is distributed in temperate regions of Europe, North America, and Western Asia [1]. *C. oxyacantha* is primarily composed of polyphenols, flavonoids, and tannins [2]. Extracts from different parts of the *C. oxyacantha* plant, such as the leaf, stem, bark, and fruit, have been used to treat different ailments.

Several studies carried out in recent decades describe the anti-inflammatory [3,4,5], gastroprotective [4,5], antimicrobial [4,5,6,7,8], hypolipidemic [5,9,10,11,12], immunomodulatory [5,13,14], antioxidant [2,5,15,16,17,18,19], and cardioprotective [2,3,5,20,21,22,23,24,25,26,27] properties of *C. oxyacantha*.

The use of medicinal plants has increased due to its effectiveness in different pathologies. The World Health Organization (WHO) approved the use of traditional medicine based on scientific evidence to demonstrate its safety and effectiveness. Approximately 80% of the world population resorts to the use of medicinal plants; however, there are many gaps in the scientific knowledge of their chemical components, mechanisms of action, and their safety [28,29,30,31].

Although *C. oxyacantha* is considered a plant with great therapeutic potential that has been used since the end of the 19th century, its safety has not been fully evaluated, in vivo and in vitro studies of *C. oxyacantha* have only reported the safety of the fruit [32,33], so this information is unknown for leaves and bark

The median lethal dose (LC_50_) of the aqueous extract of *C. oxyacantha* leaf (13.5 g/kg) and the genotoxic effects of the hydroalcoholic fruit extract at doses of 50, 100, and 200 mg/kg in bone-marrow leukocytes of mouse have been established by using the micronucleus assay [34]. Genotoxic effects have also been established in human cell cultures at doses greater than 100 µg/mL of the hydroalcoholic fruit extract of *C. oxyacantha* [35].

To determine the safety of the medicinal plant *C. oxyacantha*, toxicological studies must be carried out, which should not only be limited to tests for acute, subchronic, or chronic toxicity, but must also include tests for cytotoxicity, genotoxicity, carcinogenicity, and teratogenicity to detect possible risks of genetic damage, oncology, or on the reproduction and development of their offspring [36,37].

Among the tests that evaluate the cytotoxicity and genotoxicity of a compound in vivo is the mouse peripheral blood micronucleus (MNs) test. MNs are chromosomal fragments or complete chromosomes that can be excluded from the nucleus spontaneously or due to clastogenic or aneu-ploidogenic agents during the mitosis process. These fragments can be differentiated by the size of the MN and the presence of centromeres [38,39,40].

However, to the best of our knowledge and belief, to date, there are no published studies on the safety of the leaf and bark of *C. oxyacantha*; therefore, in this study, the possible genotoxic and cytotoxic effects of the aqueous and hydroalcoholic extracts of the bark and leaves of *C. oxyacantha* were evaluated.

## 2. Results

The cytotoxic and genotoxic results of the aqueous and hydroalcoholic leaf and bark extracts of *C. oxyacantha* in a murine model are presented in Figure 1, Figure 2 and Figure 3.

The negative control group did not present significant changes in the proportion of polychromatic erythrocytes (PCEs), micronucleated polychromatic erythrocytes (MNPCEs), and micronucleated erythrocytes (MNEs), with respect to its baseline value (0 h).

In contrast, the positive control group (PC) showed significant decreases in the proportion of PCEs at 48, 72, 96, and 120 h (Figure 1) and a statistically increased proportion of MNPCEs from 24 to 120 h (Figure 2), with the highest increase at 48 h, including the proportion of MNEs from 48 to 120 h (Figure 3) in relation to its baseline value.

In regard to the evaluated doses of aqueous leaf extract of *C. oxyacantha* (Figure 1A), only the high dose (2000 mg/kg) showed a statistically significant decrease in the proportion of PCEs at 72 h with respect to its baseline value (*p*-value = 0.005). 

The doses of 2000 and 1000 mg/kg of the hydroalcoholic leaf extract of *C. oxyacantha* showed the greatest decrease in the proportion of PCEs at 120 h; however, this difference was not statistically significant (*p*-value = 0.058). The 500 mg/kg dose also showed a non-significant decrease in this proportion, which was more evident at 96–120 h.

Regarding the evaluated doses according to the different treatment schedules, the aqueous and hydroalcoholic bark extract of *C. oxyacantha* (Figure 1B) did not show a statistically significant decrease in the proportion of PCEs.

The study groups that received the different doses of the aqueous and hydroalcoholic leaf extracts of *C. oxyacantha* and those that received the hydroalcoholic bark extract did not show a significant increase in the proportion of MNPCEs in the evaluated time schedules (Figure 2); only the aqueous bark extract of *C. oxyacantha* at the dose of 1000 mg/kg showed a statistical increase in this proportion at 120 h (*p*-value = 0.043) (Figure 2B).

The groups that received the 2000 mg/kg dose of the aqueous and hydroalcoholic leaf and bark extracts of *C. oxyacantha* showed a statistically significant increase in the proportion of MNEs (Figure 3A,B). Similarly, the dose of 1000 mg/kg of the hydroalcoholic leaf extract of *C. oxyacantha* showed a statistical increase in relation to the basal value of this proportion at 96 and 120 h (*p*-value = 0.005 and 0.011, respectively) (Figure 3B).

## 3. Discussion

The therapeutic and toxicological properties that plants possess are due to the great diversity of secondary metabolites that they may contain, as well as the concentration in which they are found. 

The genotoxicity of aqueous and hydroalcoholic leaf and bark extracts of *C. oxyacantha* was determined by increasing the number of MNEs and MNPCEs. An increase in the number of MNPCEs indicates damage occurring during 24–48 h, while the increase in the number of MNEs indicates the accumulated damage that will be observed during 96–120 h after administration [41].

The cytotoxicity was evaluated by decreasing the number of PCEs. The proportion of PCEs in the blood circulation is constant, but it can be altered by receiving a cytotoxic compound; therefore, the value of the proportion can drop to zero, indicating myelosuppression [42].

In this study, CP was used as a positive control, since it is a clastogenic alkylating compound chemically related to nitrogen mustard. Due to its carcinogenic, mutagenic, and teratogenic effects, in addition to being a known micronucleogenic agent, it was used as a positive control in genotoxicity testing [43,44,45].

As expected, the group that received CP (60 mg/kg) showed cytotoxic and genotoxic effects in the short- and long-term.

The aqueous leaf extract of *C. oxyacantha* at the dose of 2000 mg/kg exhibited cytotoxic and long-term genotoxic effects, and the hydroalcoholic leaf extract of *C. oxyacantha* at the doses of 1000 and 2000 mg/kg also showed long-term genotoxic effects.

The aqueous bark extract of *C. oxyacantha* presented short-term genotoxicity at doses of 1000 mg/kg, and at 2000 mg/kg, a long-term genotoxic effect was observed. The hydroalcoholic bark extract of *C. oxyacantha* only showed genotoxic long-term effects at the dose of 2000 mg/kg.

No published safety studies of aqueous and hydroalcoholic leaf and bark extracts of *C. oxyacantha* were found to compare our results. However, the safety of *C. oxyacantha* fruit has been described. 

De Quadros et al. evaluated the cytotoxic, genotoxic, clastogenic, and/or aneugenic damage of the hydroalcoholic fruit extract of *C. oxyacantha* in vitro by means of the cytokinesis-block micronucleus (CBMN) method. Results showed that 10 µg/mL of extract increased the number of binucleated cells, and a genotoxic effect at concentrations above 5 µg/mL was reported based on the data produced by the comet assay. In addition, cytotoxic damage in human cells (leukocytes and hepatocytes) and in *Salmonella* strains TA98 and TA100 [35] was also shown. 

Yonekubo et al. evaluated the cytogenetic damage of the hydroalcoholic fruit extract of *C. oxyacantha* at doses of 50, 100, and 200 mg/kg in mouse bone-marrow leukocytes, using the MN test. The results showed an increase in the number of MNPCEs, which translated into a short-term genotoxic effect caused by clastogenic and/or aneugenic damage, as well as a non-significant decrease in the number of PCEs. Likewise, they evaluated the direct damage to deoxyribonucleic acid, using the comet assay, in which no significant alterations were observed in the total number of cells with DNA damage [34]. Our results differ from these findings, since the hydroalcoholic leaf and bark extracts of *C. oxyacantha* did not show a significant increase in the number of MNEPCs in mouse peripheral blood, and only the aqueous bark extract at the dose of 1000 mg/kg showed a significant increase in this proportion.

These differences may be due to the fact that secondary metabolites are present in leaves, bark, and fruit, and there is a greater number of flavonoids in the leaves than in fruits (flavonol glycosides and flavones) that provide protection against UV rays, since these accumulate in the epidermal cells of the leaf. On the other hand, the fruit contains a higher concentration of anthocyanins and procyanidins than the leaves, since they are responsible for the flowering process and the colors of the fruit [7,46].

There are few reports that evaluate the genotoxic and cytotoxic potential of the different compounds that are present in the leaf of *C. oxyacantha*. For example, Bhalli, et al. evaluated the genotoxic potential of caffeic acid in vivo through the micronucleus assay and the comet assay. Results showed that, through the MN assay, caffeic acid generated a genotoxic effect by increasing the number of micronucleated cells; however, the comet assay did not demonstrate a significant increase in the number of cells with DNA damage [47]. *C. oxyacantha* leaves contain vitexin, isovitexin, and orientin. Mohammed et al. demonstrated the cytotoxic effect of these compounds extracted from the leaves of *Gleditsia tricanthos* L. in different cell lines of liver, breast, and colon cancer [48].

Tabach et al. evaluated the aerial parts of *C. oxyacantha* (26.7%), *Passiflora incarnata* (33.3%), and *Valeriana officinalis* (40%) through the Ames and micronucleus tests, concluding that the association of three different plants did not potentiate the toxic effects that could be attributed to each one separately [32].

## 4. Materials and Methods

### 4.1. Materials and Reagents

For general procedures, the reagents employed were reactive grade of the commercial brands J. T Baker (Mexico) and Golden Bell (Mexico). Cyclophosphamide (CAS 6055-19-2) and acridine orange (CAS 10127-02-3) were from Sigma-Aldrich (St. Louis, MO, USA).

### 4.2. Plant Material

The leaves and bark of *C. oxyacantha* were obtained from the supplier Nutra Herbal de Mexico (Convento de Balvanera #24 Col. Jardines de Santa Monica, Mexico, Tlalnepantla C.P. 54050, Mexico).

### 4.3. Preparation of the Aqueous and Hydroalcoholic Leaf and Bark Extracts of C. oxyacantha

The dried leaves and bark of the plants were ground to a fine powder with a particle size of less than 0.5 mm. The aqueous extraction was carried out by using the decoction method. The powder was dissolved in a ratio of 1 g per 10 mL of water, boiled for 15 min, then filtered to eliminate the plan, and the resulting solution was frozen and lyophilized.

The preparation of the hydroalcoholic leaf and bark extracts of *C. oxyacantha* was carried out by mechanical maceration in a proportion of 10 g of the plant in 300 mL of 70% ethanol for 48 h.

The mixture was refluxed (62 °C) for 2 h and filtered. Activated carbon was added to the resulting filtrate and it was mechanically macerated for 48 h to remove pigments before filtering once again. Ethanol was removed on a rotary evaporator and subsequently lyophilized.

### 4.4. Animals

Seventy (n = 70) 3-month-old adult male clinically healthy Balb-C mice were placed into polycarbonate cages with food and tap water (Harlan Teklad Lab Blocks) ad libitum. The animals were provided by the Claude Bernard Animal Facility at the Health Sciences Area UAZ-Siglo XXI from the Autonomous University of Zacatecas. The project was approved by the Ethics and Research Committee of the Autonomous University of Zacatecas (registration number ACS/UAZ/051/2010).

### 4.5. Groups and Test Agent Treatments

Fourteen experimental groups of 5 mice each were distributed as follows: Group 1 received sterile water (negative control); Group 2, 60 mg/kg of cyclophosphamide (CP) divided into two doses (positive control); Groups 3–8, one of the following doses (0.5, 1, and 2 g/kg) of aqueous and hydroalcoholic leaf extract of *C. oxyacantha*; and Groups 9–14 were administered one of the following doses (0.5, 1 and 2 g/kg) of aqueous and hydroalcoholic bark extract of *C. oxyacantha*. All doses were administered orally with an esophageal cannula for 5 days, and the volume of administration was 0.1 mL/10 g of weight.

### 4.6. Sample Preparation and Micronucleus Analysis in Mice

The evaluation of genotoxic and cytotoxic damage was determined by the micronucleus test (MNs) [38]. A drop of peripheral blood was obtained at 0, 24, 48, 72, 96, and 120 h after the administration of the corresponding dose from the tail tip of mice in each group and two smears from each mouse were made on previously encoded clean and degreased slides. The smears were fixed in absolute ethanol for 10 min and stained with acridine orange [49]. An Olympus CX31 microscope equipped with epifluorescence and an oil immersion objective (100x) was used to evaluate the genotoxic and cytotoxic damage. The number of polychromatic erythrocytes (PCEs) was counted in 1000 total erythrocytes (TEs), the number of micronucleated polychromatic erythrocytes (MNPCEs) in 1000 PCEs, and the number of micronucleated erythrocytes (MNEs) in 10,000 TEs.

### 4.7. Statistical Analysis

Data were expressed as mean ± standard deviation per group. The comparison was performed between each group and its respective baseline value (0 h), using the analysis of variance (ANOVA) for repeated measures. The Bonferroni adjustment test was used for multiple post hoc comparisons. Statistical significance was set at *p* < 0.05. Data analysis was performed by using IBM SPSS (V25) statistics program for Windows.

### 4.8. Ethical Considerations

The animals were treated according to the regulations and the official standards in force, such as the official Mexican norm NOM-062-ZOO-1999, which shows the specifications and techniques for the production, care and use of institutional laboratory animals. Furthermore, animals were sacrificed according to the NOM-033-SAG/ZOO-2014 and their remains were treated as indicated by the NOM-087-ECOL-SSA1-2002 for Environmental Protection-Environmental Health-Infectious Biological Hazardous Waste-Classification and Handling Specifications.

## 5. Conclusions

Under the experimental conditions used in this study, the aqueous and hydroalcoholic leaf and bark extracts of *C. oxyacantha* showed long-term genotoxic damage at the highest tested dose, and only the aqueous bark extract of *C. oxyacantha* showed short-term genotoxicity at the average tested dose. In relation to cytotoxic activity, only the highest tested dose of the aqueous leaf extract of *C. oxyacantha* showed this effect. Based on the results, the hydroalcoholic and aqueous extracts of the leaf and bark of *C. oxyacantha* are safe at doses less than 1000 mg/kg.

## Figures and Tables

**Figure 1 plants-10-02217-f001:**
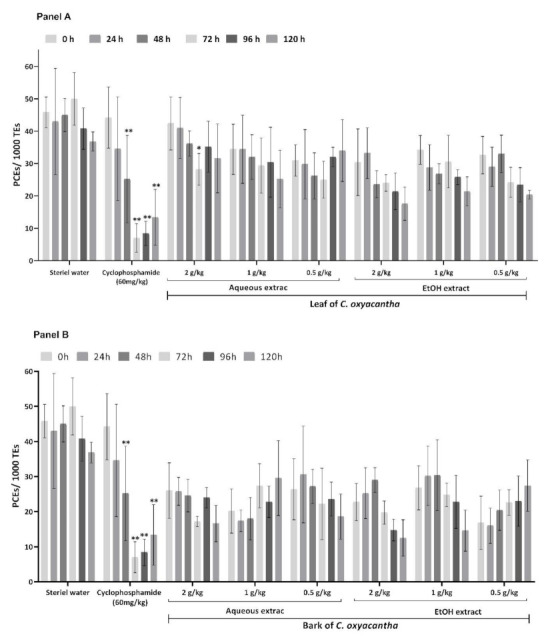
Proportion of PCEs in mouse peripheral blood in the study groups. Panel **A** shows the results obtained from the leaf and panel **B** those from the bark of *C. oxyacantha* Intragroup comparisons were performed between baseline samples (0 h) against the following sampling times: 24, 48, 72, 96, and 120 h. PCEs, Polychromatic Erythrocytes; * *p* < 0.05; ** *p* < 0.001.

**Figure 2 plants-10-02217-f002:**
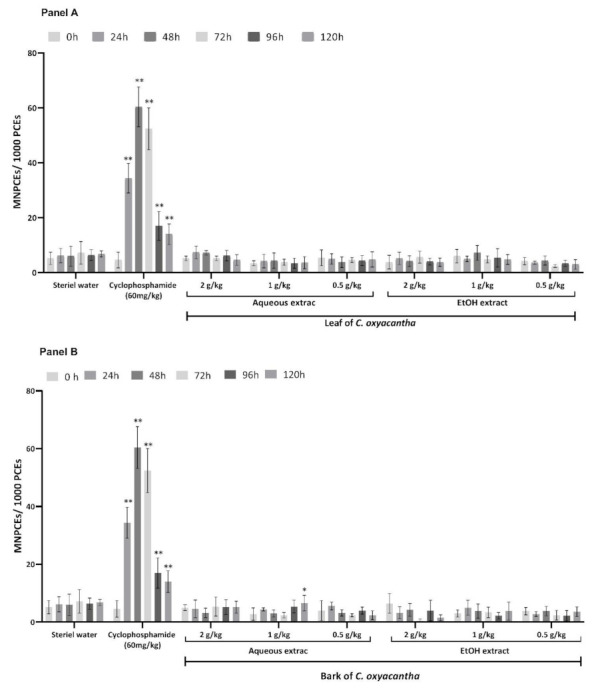
Proportion of MNPCEs in mouse peripheral blood in the study groups. Panel **A** shows the results obtained from the leaf and panel **B** those from the bark of *C. oxyacantha*. Intragroup comparisons were performed between baseline samples (0 h) against the following sampling times: 24, 48, 72, 96, and 120 h. MNPCsE, Micronucleus Polychromatic Erythrocytes; * *p* < 0.05; ** *p* < 0.001.

**Figure 3 plants-10-02217-f003:**
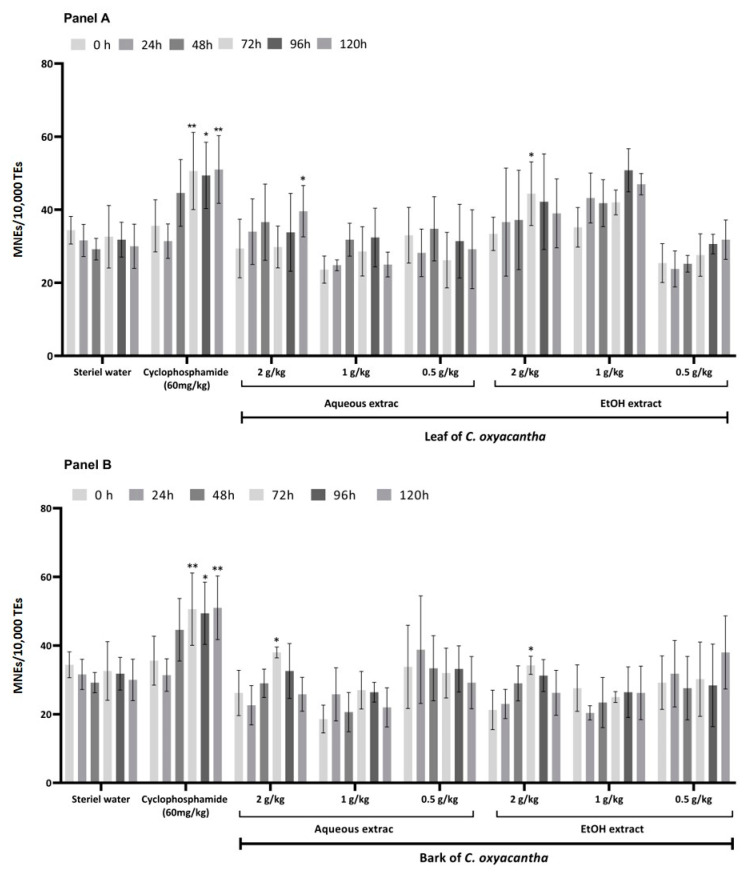
Proportion of MNEs in mouse peripheral blood in the study groups. Panel **A** shows the results obtained from the leaf and panel **B** those from the bark of *C. oxyacantha.* Intragroup comparisons were performed between baseline samples (0 h) against the following sampling times: 24, 48, 72, 96, and 120 h. MNEs, Micronucleus Erythrocytes; * *p* < 0.05; ** *p* < 0.001.

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
