# Peer review of "Cytotoxic and Genotoxic Evaluation of the Aqueous and Hydroalcoholic Leaf and Bark Extracts of Crataegus oxyacantha in Murine Model"

_plants, 2021, doi:10.3390/plants10102217_

Round 1
Reviewer 1 Report
Article : "Cytotoxic and genotoxic evaluation of the aqueous and hydroalcoholic leaf and bark extracts of Crataegus oxyacantha in murine model ".
- Important taxonomic data of the plant lack; e.g. taxonomic author name, family name of the species.
- The article deals with simple preliminary chemical tests for identification of main chemical groups in the plant. While many pure polyphenols and tannins were previously isolated from the bark and leaf of the plant [Sokół-Łętowska; Oszmianski; Benabderrahmane; Svedström]. Therefore, the result of the chemical content is very known or less than the previous one.
Author Response
Response to Reviewer 1 Comments
We appreciate your attention to our manuscript entitled “Cytotoxic and genotoxic evaluation of the aqueous and hydroalcoholic leaf and bark extracts of Crataegus oxyacantha in murine model". The suggestion that you gently made on the paper, were made as follow:
- Important taxonomic data of the plant lack, e.g. taxonomic author name, family name of the species.
Our response: We change the first introductory paragraph to:
Crataegus oxyacantha is a deciduous tree commonly known as white hawthorn, is a member of Rosaceae family, Magnoliopsida clase, Rosales order, Plantae kingdom and Magnoliophyta division, it is distributed in temperate regions of Europe, North America, and Western Asia
- The article deals with simple preliminary chemical tests for identification of main chemical groups in the plant. While many pure polyphenols and tannins were previously isolated from the bark and leaf of the plant [Sokół-Łętowska; Oszmianski; Benabderrahmane; Svedström]. Therefore, the result of the chemical content is very known or less than the previous one.
Our answer:
We agree that the chemical content of C. oxyacantha already established, so the main objective of the article was not to determine the chemical composition of the extracts, but to establish their safety. Nevertheless, we wanted to point that according to the type of extraction, are the chemical compounds that are obtained the plant.
However, based on your comment, we decided to remove that part and focus on the main objective of the article, which is the security of the excerpts.
Reviewer 2 Report
Dear Authors,
Crataegus oxycantha has been widely explored and many references can be found in the literature. Nevertheless, any further study of this plant is considered interesting. However, in my opinion, the study presented here lacks important scientific sounds. In addition, the format used is incorrect according to the journal.
Relevant considerations:
-This year is 2021, the chemical study of plants and other organisms is accompanied by a plethora of analytical and spectroscopic methods, so you cannot say in line 73 "In this study, the major secondary metabolites were identified by phytochemical analysis...". In this study, a qualitative chemical study was performed using generalist methods.
-The information appears repeated. Table 2 gives the same information as shown in Figures 1, 2, and 3. The table or figures should be in the supplementary material.
-Discussion. It is not presented accurately and instead of a scientific discussion, the authors present a short review that could be better placed in the introduction.
-Conclusion. As far as I know, the objective of this study was to correlate the chemical study with the genotoxic and cytotoxic effect of Crataegus oxycantha, however, as the chemical study was brief, the conclusions are sparse and non-diagnostic.
-Although the writing style may be considered understandable, however, it is not always clear and I suggest careful reading, some improvements throughout the manuscript, and perhaps external revision.
Minor considerations:
-plant name always in italics.
-references in the text with numbers.
-mL (sometimes ml appears)
-LC50
-line 50: 100 micrograms ml?
With all these considerations, I urge the authors to take into account the above suggestions and rewrite the manuscript according to the results obtained.
Author Response
Response to Reviewer 2 Comments
We appreciate your attention to our manuscript entitled “Cytotoxic and genotoxic evaluation of the aqueous and hydroalcoholic leaf and bark extracts of Crataegus oxyacantha in murine model" The suggestion that you gently made on the paper, were made as follow:
Crataegus oxycantha has been widely explored and many references can be found in the literature. Nevertheless, any further study of this plant is considered interesting. However, in my opinion, the study presented here lacks important scientific sounds. In addition, the format used is incorrect according to the journal.
Relevant considerations:
-This year is 2021, the chemical study of plants and other organisms is accompanied by a plethora of analytical and spectroscopic methods, so you cannot say in line 73 "In this study, the major secondary metabolites were identified by phytochemical analysis...". In this study, a qualitative chemical study was performed using generalist methods.
Our answer: We agree that the chemical content of C. oxyacantha already established, so the main objective of the article was not to determine the chemical composition of the extracts, but to establish their safety. Nevertheless, we wanted to point that according to the type of extraction, are the chemical compounds that are obtained the plant.
However, based on your comment, we decided to remove that part and focus on the main objective of the article, which is the security of the excerpts.
-The information appears repeated. Table 2 gives the same information as shown in Figures 1, 2, and 3. The table or figures should be in the supplementary material.
Our answer: The Table 2 will be transferred to supplementary material and only the figures 1, 2 and 3 were left alone were left in the results section
-Discussion. It is not presented accurately and instead of a scientific discussion; the authors present a short review that could be better placed in the introduction.
Our answer: The short review of the discussion was removed, and the introduction was completed with it
-Conclusion. As far as I know, the objective of this study was to correlate the chemical study with the genotoxic and cytotoxic effect of Crataegus oxycantha, however, as the chemical study was brief, the conclusions are sparse and non-diagnostic.
Our answer: The objective of the study was not to correlate the chemical study with the genotoxic and cytotoxic effect of Crataegus oxycantha. The main objective was to evaluate the safety of the aqueous and hydroalcoholic extracts of leaves and bark of C. oxyacantha, by means of the in vivo micronucleus test. For this reason, eliminating the results from the physicochemical part will eliminate confusion
-Although the writing style may be considered understandable, however, it is not always clear and I suggest careful reading, some improvements throughout the manuscript, and perhaps external revision.
Our answer: We made an improvement in writing and the manuscript was edited by a professional in the area
Minor considerations:
-plant name always in italics.
Our answer: all plant names were changed to italics
-references in the text with numbers.
Our answer: all references in the text were changed to numbers
-mL (sometimes ml appears)
Our answer: all the words (ml) were changed by (mL)
-LC50
Our answer: The median lethal dose (LD50) was changed by (LC50)
-line 50: 100 micrograms ml?
Our answer: 100 micrograms ml was changed by 100 µg/mL
With all these considerations, I urge the authors to take into account the above suggestions and rewrite the manuscript according to the results obtained.
Our answer: All changes suggested by the reviewer were made
Reviewer 3 Report
Dear Authors,
There is an interesting and original idea in your manuscript "Cytotoxic and genotoxic evaluation of the aqueous and hydroalcoholic leaf and bark extracts of Crataegus oxyacantha in murine model", which you have sent for publication.
Introduction
The well-written introduction is supported by the relevant quotations.
Results
The results are presented well and clearly.
Methods and materials
The authors have well described the methods they use and could easily be repeated by researchers. The methods used for phytochemical analysis have long been known, easy and simple, which allows the reader of this manuscript to quickly navigate which of the studied phytochemicals. compound are found in the aqueous and hydroethanol extracts.
Discussion
According to the phytochemical analysis, the leaf of C. oxyacantha contains flavonoles, chalcones, tannins such as catechins and derivatives of galic acid, and derivatives of anthrone.
I agree with this paragraph. With the next paragraph I would like the authors to answer me how they managed to prove the presence of the discussed compounds after using only a phytochemical test? Do the authors prove the presence of the discussed phytochemical compounds in the bark only from literature data?
The phytochemical profile of the bark of C. oxyacantha indicates the presence of aurons, chalcones, flavonoles, gallic acid derivatives and anthraquinones.
Conclusion
Nothing about phytochemicals was noted in the conclusion. Do they have any effect on the results obtained?
Author Response
Response to Reviewer 3 Comments
We appreciate your attention to our manuscript entitled “Cytotoxic and genotoxic evaluation of the aqueous and hydroalcoholic leaf and bark extracts of Crataegus oxyacantha in murine model". The suggestion that you gently made on the paper, were made as follow:
Dear Authors,
There is an interesting and original idea in your manuscript "Cytotoxic and genotoxic evaluation of the aqueous and hydroalcoholic leaf and bark extracts of Crataegus oxyacantha in murine model", which you have sent for publication.
Introduction
The well-written introduction is supported by the relevant quotations.
Results
The results are presented well and clearly.
Methods and materials
The authors have well described the methods they use and could easily be repeated by researchers. The methods used for phytochemical analysis have long been known, easy and simple, which allows the reader of this manuscript to quickly navigate which of the studied phytochemicals. compound are found in the aqueous and hydroethanol extracts.
Our answer: the section on the phytochemical analysis methodology was eliminated based on the controversy generated over which was the objective of the work
Discussion
According to the phytochemical analysis, the leaf of C. oxyacantha contains flavonoles, chalcones, tannins such as catechins and derivatives of galic acid, and derivatives of anthrone.
I agree with this paragraph. With the next paragraph I would like the authors to answer me how they managed to prove the presence of the discussed compounds after using only a phytochemical test? Do the authors prove the presence of the discussed phytochemical compounds in the bark only from literature data?
The phytochemical profile of the bark of C. oxyacantha indicates the presence of aurons, chalcones, flavonoles, gallic acid derivatives and anthraquinones.
Our answer: There are already finer reports (HPLC, NMR) that establishing the main components of the leaf, bark and fruit of C. oxyacantha, however, the objective of presenting the phytochemical results was only to establish that depending on the part of the plant and type of solvent used for the extraction, the chemical compounds can vary, however, it was not our main objective to determine the chemical compounds of the plant and given the controversy that occurred we decided to eliminate those results, since the main objective was establish the safety of the aqueous and hydroalcoholic extracts of the leaf and bark of C. oxyacantha in vivo
Conclusion
Nothing about phytochemicals was noted in the conclusion. Do they have any effect on the results obtained?
Our answer: The amount and types of chemical components obtained depend on the part of the plant used as well as the type of solvent used for its extraction, therefore, if it can affect the results obtained. In the discussion section describes what has been published on this topic (line 168-182). Nevertheless, the objective of the work was only determine the cytotoxicity and genotoxicity the aqueous and hydroalcoholic leaf and bark extracts of C.oxyacantha in murine model, in order to evaluate its safety.
Round 2
Reviewer 1 Report
Article: (Cytotoxic and genotoxic evaluation of the aqueous and hydroalcoholic leaf and bark extracts of Crataegus oxyacantha in murine model)
- Line 37: it is enough to write the name of the plant family. Author name of the taxonomic identification lacks; it is very important by the writing of the botanical name of the plant species.
- Line: "clase" ?!; correct the language across the manuscript
- Writing format of the references must be corrected and improved.
Author Response
We appreciate your comments to the article "Cytotoxic and genotoxic evaluation of the aqueous and hydroalcoholic leaf and bark extracts of Crataegus oxyacantha in murine model"
- Line 37: it is enough to write the name of the plant family. Author name of the taxonomic identification lacks; it is very important by the writing of the botanical name of the plant species.
Our response: we changed line 36 and 37 based on your suggestion to:
Crataegus oxyacantha ssp monogyna (hawthorn) is a deciduous tree commonly known as white hawthorn, is a member of Rosaceae family
- Line: "clase" ?!; correct the language across the manuscript
Our response: The manuscript was edited by a professional in the area (the certificated is attached)
- Writing format of the references must be corrected and improved.
Our response: the reference format was corrected and standardized

Reviewer 2 Report
Dear authors,
In this new version, it is well established the main objective of this work and removing any consideration about the phytochemical analysis has led to improve it.
Author Response
We appreciate the comments and recommendations made to the article Cytotoxic and genotoxic evaluation of the aqueous and hydroalcoholic extracts of leaves and bark of Crataegus oxyacantha, with which we were able to establish the main objective of the work
Reviewer 3 Report
Dear Authors,
I agree with the corrections you have made to your manuscript. You have answered my questions and I agree that your manuscript be accepted.
Author Response
I agree with the corrections you have made to your manuscript. You have answered my questions and I agree that your manuscript be accepted.
Our response: We appreciate the comments and recommendations made to the article Cytotoxic and genotoxic evaluation of the aqueous and hydroalcoholic leaf and bark extracts of Crataegus oxyacantha, which were very important to improve the article.
Round 3
Reviewer 1 Report
Article: Cytotoxic and genotoxic evaluation of the aqueous and hydroalcoholic leaf and bark extracts of Crataegus oxyacantha in murine model.
Line 35: "Crataegus oxyacantha ssp monogyna (hawthorn)"
- The person/s making the original description in a published journal or book is called the 'author' for that plant name, and their name follows the genus and species in a full citation, for example: Acacia nilotica (L.) Delile; where (L.) Delile is the author of the plant name. Also, what is the author for the name of your plant species.
- (hawthorn) is written not in italic format
Line 196: "The leaves and bark of C. oxyacantha were obtained from the supplier Nutra Herbal de Mexico (Convento de Balvanera Col. Jardines de Santa Monica, Mexico, Tlal nepantla C.P. 54050, Mexico)".
How could you identified the taxonomic name for the plant, where and who made this taxonomic identification.
Author Response
Response to Comments and Suggestions for Author
Article: Cytotoxic and genotoxic evaluation of the aqueous and hydroalcoholic leaf and bark extracts of Crataegus oxyacantha in murine model.
Line 35: "Crataegus oxyacantha ssp monogyna (hawthorn)"
- The person/s making the original description in a published journal or book is called the 'author' for that plant name, and their name follows the genus and species in a full citation, for example: Acacia nilotica (L.) Delile; where (L.) Delile is the author of the plant name. Also, what is the author for the name of your plant species.
Answer:
Line 35: "Crataegus oxyacantha ssp monogyna (hawthorn)" was changed by " Crataegus oxyacantha L. (Rosaceae)
- (hawthorn) is written not in italic format
Answer: was eliminated of line 35 the word hawthorne
Line 196: "The leaves and bark of C. oxyacantha were obtained from the supplier Nutra Herbal de Mexico (Convento de Balvanera Col. Jardines de Santa Monica, Mexico, Tlal nepantla C.P. 54050, Mexico)".
How could you identified the taxonomic name for the plant, where and who made this taxonomic identification.
Answer: The plant supplier, Nutra Herbal de México, is in charge of taxonomic identification. The supplier sends the taxonomic information of the plant that is purchased.
In the case of the plant that we buy, I send the following information:
Date of purchase: January 22, 2014
Scientific name: Crataegus oxyacantha L.
Kingdom: Plantae
Division: Magnoliophyta
Class: Magnoliopsida
Order: Rosales
Family: Rosaceae
Species: Crataegus monogyna
Round 4
Reviewer 1 Report
Line 258: methods for taxonomically identification of the plant species is not clear. Specimen for the plant species must be identified taxonomic by a botanist and plant specimen with specimen voucher must collected and deposited at an official research center.
Many writing format must be corrected across the manuscript, examples"
Line 39: (Rosaceae), written not in italic format
Line 65: (in vivo); (in vitro), check in italic form to write
Author Response
We appreciate all your suggestions and questions
- Line 258: methods for taxonomically identification of the plant species is not clear. Specimen for the plant species must be identified taxonomic by a botanist and plant specimen with specimen voucher must collected and deposited at an official research center.
Answer
The plant was acquired from a company specialized in plant, which supervises the taxonomic identification of the plant and sends us a sheet with the taxonomic specifications of the plant.
C. oxyacantha is not originally from Mexico, so it was purchased from a supplier.
When it comes to a plant native to our country, the collection of the plant is carried out by us together with one of the biologists which is part of the Herbarium of the State of Zacatecas.
The biologist Emmeth J. Rodríguez Pérez, oversees the taxonomic identification of the plant and a specimen is kept in the Herbarium of the State of Zacatecas
In section 4.2 Plant material, you will find the data of the company with which the plant species was acquired and oversaw the taxonomic identification.
- Many writing format must be corrected across the manuscript, examples"
Line 39: (Rosaceae), written not in italic format
Line 65: (in vivo); (in vitro), check in italic form to write
Answer
Writing formats were corrected throughout the manuscript, words in vitro and in vivo were italicized and Rosaceae italics removed